# Neural representations of situations and mental states are composed of sums of representations of the actions they afford

**Mark A. Thornton** [1] ✉ **& Diana I. Tamir** [2,3]

Human behavior depends on both internal and external factors. Internally, people's mental states motivate and govern their behavior. Externally, one's situation constrains which actions are appropriate or possible. To predict others' behavior, one must understand the influences of mental states and situations on actions. On this basis, we hypothesize that people represent situations and states in terms of associated actions. To test this, we use functional neuroimaging to estimate neural activity patterns associated with situations, mental states, and actions. We compute sums of the action patterns, weighted by how often each action occurs in each situation and state. We find that these summed action patterns reconstructed the corresponding situation and state patterns. These results suggest that neural representations of situations and mental states are composed of sums of their action affordances. Summed action representations thus offer a biological mechanism by which people can predict actions given internal and external factors.

Humans perform an extensive variety of actions on a regular basis. Which of these actions a person chooses to perform at any given moment depends on both external and internal factors[1-6]. The external, or exogenous, factors that shape a person's behavior comprise a person's situation. Situations comprise collections of physical and social factors that constrain the behaviors that are appropriate – or indeed, possible – for a person to perform. The internal, or endogenous, factors that shape behavior include people's mental states. These states include the moods, emotions, and desires which motivate actions, and the reasoning, calculating, and planning states that help to select and govern actions. Anticipating which actions others are likely to perform is critical for navigating everyday social interactions. Making accurate predictions requires people to understand both the internal and external influences on others' behavior. Here we investigated whether the brain represents situations and mental states in a way that facilitates social prediction. Drawing inspiration from both ecological psychology and the theory of predictive coding[1,3,7-9], we hypothesize that people represent situations and mental states as weighted sums of associated actions. We test this hypothesis here

using a combination of functional magnetic resonance imaging (fMRI), behavior, and text analysis.

Situations are among the most important shapers of human behavior. Situations refer to the clusters of norms, schemas, and scripts that help people easily navigate stereotyped interactions, such as having dinner at a restaurant[10,11]. Situations also include the physical characteristics of the environment, such as which objects are present, that constrain the set of actions people could undertake[1,12]. Situational schema and norms tell us when and where certain actions are appropriate. For example, people know to dance at (most) weddings, but not at (most) funerals. Sometimes a situation's affordances are binary: one simply cannot check email without an internet connection. Other affordances are graded: nearby jackhammering does not completely rule out sleep, but it does make it considerably less likely. Scripts inform the sequences of behaviors people execute in each situation: buy a ticket, get popcorn, sit down, and then watch the movie; not the reverse order. Strong social situations can even cause people to act in uncharacteristic ways, such as ignoring someone in need when one is late to a meeting[13] or harming someone on an authority figure's orders[14].

[1]Department of Psychological and Brain Sciences, Dartmouth College, Hanover, NH 03755, USA. [2]Department of Psychology, Princeton University, Princeton, NJ 08540, USA. [3]Princeton Neuroscience Institute, Princeton University, Princeton, NJ 08540, USA. ✉e-mail: Mark.a.thornton@dartmouth.edu

Mental states are the primary endogenous shaper of human behavior. Mental states comprise the affective states (e.g., emotions, such as happiness or envy) and the cognitive states (such as planning, calculation, and decision-making) that motivate and govern behavior. Certain mental states promote certain actions. For instance, concern often leads to helping behavior, whereas anger promotes aggression[15]. Mental states can also constrain behavior. An extreme emotion – such as panic – may limit one's normally rich behavioral repertoire to a rudimentary set of prepared behavior responses: fight or flight[16]. Occupying cognitive states such as planning or imagination may reveal new possibilities to act on[17], or help one transition between disparate emotions, as in cognitive reappraisal[18]. Some mental states, such as intentions, are closely tied to specific actions. Here we focus on more general cognitive and affective states without specific propositional content.

To effectively navigate social life, people must accurately predict others' behavior[9,19–21]. This is true both in cooperative contexts, in which people must coordinate on synchronized or complementary actions, and in competitive contexts, in which one must anticipate another's actions to foil their goals and achieve own's one. Given the pervasive influence of situations and mental states on actions, a social agent would benefit from understanding these shapers of the social world. What would an effective understanding of situations and mental states look like? Two distinct perspectives on the human brain – the theory of predictive coding and ecological psychology – both point to a similar answer: the brain may represent situations and states as probability-weighted sums of the actions they predict.

The theory of predictive coding suggests that prediction is the central organizing principle of the brain[8,22]. This theory first achieved prominence in explaining aspects of visual processing[23,24] but has since been successfully applied to a wide range of domains, including social cognition[9,25–27]. Its central tenant is that the brain does not merely perceive its inputs, but actively predicts them. Thus, prediction errors, rather than full input signals, ascend the cortical processing hierarchy, and ever-adjusting predictions filter back down as feedback. This principle implies that neural representations of the external world do not merely reflect the world's current state, but rather predictions about how the world is likely to evolve in the future. For instance, neural representations of visual objects reflect how frequently those objects co-occur in natural scenes: objects that often tend to co-occur elicit similar patterns of brain activity, even if they do not visually resemble each other[28]. This follows from predicting code, in that perceiving any one object results in predictions about what other objects might also be around to see.

The brain processes others' mental states in accordance with predictive coding. For instance, whenever one thinks about a mental state, the pattern of brain activity elicited systematically resembles the patterns elicited by other states that tend to follow that state in real life[27]. For example, thinking about someone feeling rage primes the brain to ponder their likely subsequent feelings of regret. Similarly, the way people conceptualize novel mental states is causally shaped by the transition dynamics between them: states which tend to precede or follow one another are judged as more conceptually similar than pairs with low transition probabilities between them[29]. If we consistently see people become calm after they think for a while, we are likely to come to conceptualize thinking and calmness similarly. Finally, the way that people represent other individuals reflects predictions about their habitual mental states. For example, in one study, participants mentalized about popular US media figures including Bill Nye (the Science Guy) and Justin Bieber (the musician)[30]. Due to his work as a science communicator, Bill Nye was frequently thought to experience the mental states of "curiosity" and "excitement" and rarely experience the mental state of "anger" and "envy." When the patterns of brain activity elicited by thinking about curiosity, excitement, anger, and envy (and other states) were summed up – weighted by how frequently Nye

experiences these states – the resulting pattern resembled the brain pattern associated with thinking about Nye himself. This summed state reconstruction is person-specific, representing Nye more than Bieber or other target people who are thought to experience mental states with different frequencies. These summed states explain which people the brain represents as similar or different to each other, over and above trait dimensions such as extraversion or competence. This work shows that mental states and identity may be related because the brain represents other people as bundles of predictions about their mental states. We propose that mental states and action representations have a similar relationship, such that the brain represents mental states as bundles of predictions about the actions those states potentiate.

Ecological psychology likewise points to the hypothesis that action predictions shape situation representations, albeit from a very different perspective. James Gibson's formative work on affordances was the first to propose that situations are defined by the actions they afford[7]. Gibson was a founder of ecological psychology, a subfield that takes a direct, embodied approach to studying perception and action[31]. Environmental affordances play a key role in explaining action within this paradigm. However, ecological psychology differs from the current mainstream in cognitive neuroscience in that it is anti-representational – i.e., it maintains that the brain does not represent the outside world. Here we restate this hypothesis from a representational perspective: just as neural representations of mental states might reflect action predictions based on internal factors, neural representations of situations might reflect action predictions based on external factors.

These disparate perspectives thus converge on the same central hypothesis: action affordances are the building blocks of situation and mental state representations. Specifically, since both situations and mental states entail specific action affordances, we hypothesize that the brain's representations of situations and states are composed of weighted sums of representations of the actions they afford. To test this hypothesis, in the present study, participants judged the likelihood of co-occurrences between situations, mental states, and actions while undergoing fMRI scanning (Fig. 1). For example, participants judged how likely it was that someone in the situation of being "in an elevator" might engage in the action "waiting" (judged to be very likely) or "bathing" (judged to be very unlikely). Within brain regions that contained reliable situation-, mental state-, and action-specific patterns of brain activity, we summed up the action patterns, weighted by their co-occurrence with each situation and each state. We found that these summed action representations specifically reconstructed the corresponding situation/state patterns. This result indicates that the brain does represent situations and mental states as sums of their action affordances.

## Results

Our main hypothesis – that actions sum to states and to situations – is predicated on neural overlap between these three types of representation. Action patterns can only sum up (voxelwise) to situation or state patterns if these patterns are found in overlapping regions. Thus, we first identified brain regions that contained reliable neural representations of situations, mental states, and actions, respectively (Fig. 2). We used reliability-based feature selection[32] to identify these respective sets of brain regions. This data-driven method identifies regions that maximize both univariate voxelwise reliability, as well as multivariate pattern similarity reliability. With optimal voxelwise reliability thresholds for mental states ($r = 0.37$), situations ($r = 0.53$), and actions ($r = 0.43$), we identified regions of maximal pattern-wise reliability for mental states ($r = 0.76$), situations ($r = 0.71$), and actions ($r = 0.43$). These reliabilities were similar in magnitude to the reliabilities observed in a previous application of reliability-based feature selection to a condition rich paradigm using social stimuli[33].

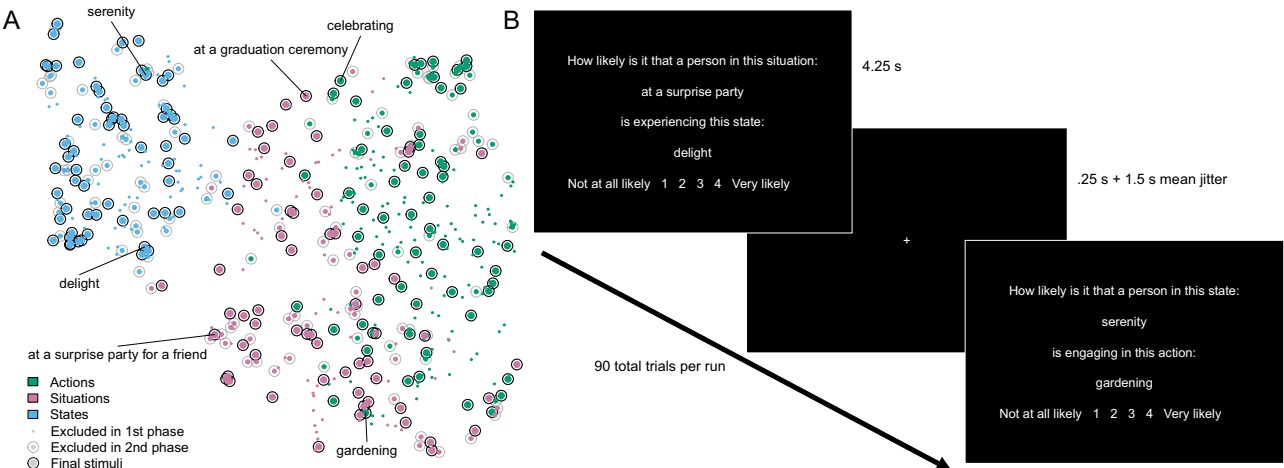

**Fig. 1 | Stimulus selection and task schematic. A** Two phases of stimulus selection were conducted to choose situations (pink), mental states (blue), and actions (green) with maximum variation in their co-occurrences, and minimal redundancy. The plot above displays all stimuli from each of the three classes. The proximity between stimuli (points) represents the similarity of their fastText embedding vectors, projected into a 2-D space via Uniform Manifold Approximation[69]. The style of the circles indicates whether stimuli were excluded during the first phase of selection, 2nd phase of selection, or retained for use in the fMRI study. Selected examples are labeled. Source data are provided as a Source Data file. **B** The schematic illustrates the structure, appearance, and timing of the task presented to participants in the fMRI scanner. Situation-action, situation-state, and state-action trials were randomly interleaved throughout each run.

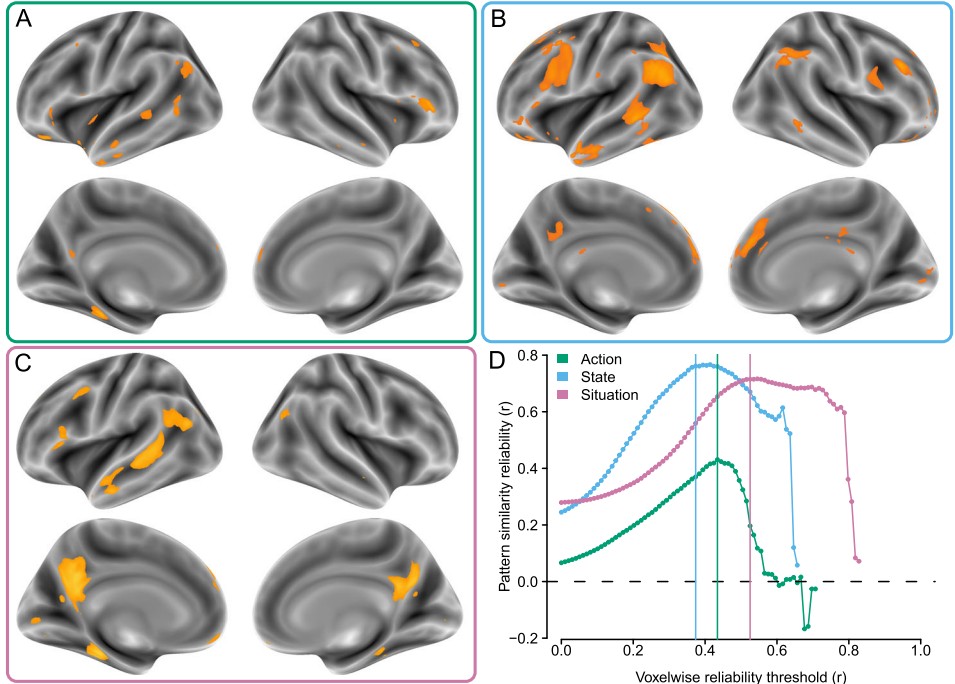

**Fig. 2 | Reliabilty-based feature selection.** Voxels involved in representing (**A**) actions (green), (**B**) mental states (blue), and (**C**) situations (pink) were identified using reliability-based feature selection (**D**). Panel (**D**) portrays the reliability of neural pattern similarity matrices for each class of stimuli, as a function of the voxelwise reliability threshold used to select voxels for inclusion in the patterns. The colored vertical lines indicate the selected thresholds, which produced the orange regions in panels (**A**–**C**). Brighter orange colors reflect higher reliability. Source data are provided as a Source Data file.

The feature selection procedure yielded disjoint sets of 5788 voxels representing situations, 10294 voxels representing mental states, and 2536 voxels representing actions (Fig. 2D). These voxels were located across multiple regions of association cortex, including the superior temporal sulcus (STS) extending from the temporoparietal junction (TPJ) to the anterior temporal lobe (ATL); medial parietal cortex, including parts of precuneus, posterior cingulate, and retrosplenial cortex; and lateral frontal cortex, including portions of premotor, ventrolateral, and dorsolateral cortex; and dorsal medial prefrontal cortex (dMPFC). In all three cases, the overall sets of regions strongly resembled the canonical social brain network, an interpretation supported by the NeuroSynth[34] Decoder tool, which listed "default [mode]" and/or "theory [of] mind" as the closest matching nonspatial terms for all three unthresholded reliability maps.

We observed regions of pairwise overlap between the voxels selected for all three classes of stimuli. A set of 570 voxels were shared between the situation and action maps, 668 voxels were shared between the mental state and action maps, and 1545 voxels were shared between the situation and mental state maps (Fig. S4). Regions of overlap across domains included the TPJ, STS, ATL, and MPFC. The

existence of this spatial overlap provides an initial indication that neural representations of situations, states, and actions may be related.

With these regions identified, we next tested our primary hypothesis that affordances shape social representations (Fig. 3). We did so by adding up patterns of one type (e.g., actions) to reconstruct patterns of another type (e.g., situations). Each sum was weighted by how often each stimulus in the former class was thought to co-occur with a given stimulus in the latter class (Fig. S1–S3). So, for example, in the situation "with an elderly relative," the action "talking" would receive a height weight in the summation, whereas the action "skiing" would receive a low weight. These pattern summation analyses corroborated two of our three preregistered confirmatory hypotheses

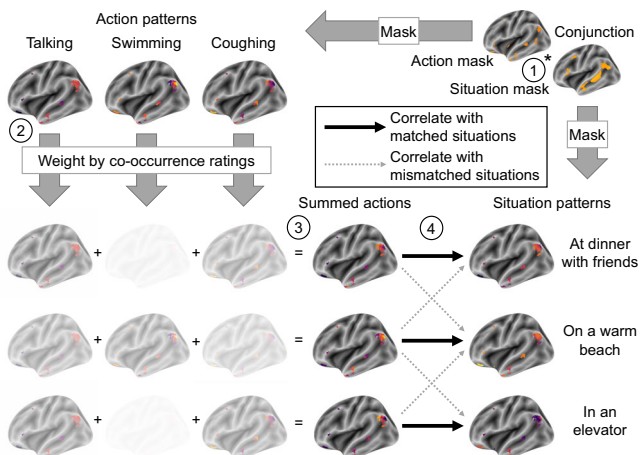

**Fig. 3 | The schematic illustrates the pattern summation analysis process.** (1) Wholebrain images associated with actions and situations were masked by the conjunction of the feature-selected masks for these two stimulus types. (2) Action patterns were weighted based on how often they were judged to co-occur with specific situations. (3) The weighted actions patterns were summed to reconstruct the corresponding situations. (4) To test the specificity of the reconstruction, we measured the correlation between summed action patterns and both the matching situation patterns and mismatched situation patterns. The difference between the two served as our accuracy metric and our primary inferential criteria (Fig. 4). This process was repeated with every pairing of situations, mental states, and actions. Colors in the brain maps reflect degree of activity in different voxels, with brighter colors indicating more activity.

(Fig. 4). Specifically, we found that frequency-weighted sums of action patterns reconstructed situation-specific patterns (mean $Z(r)$ difference $_{matched-mismatched}$ = 0.0017, $d$ = 0.49, $p_{corrected}$ = 0.043) and that that frequency-weighted sums of action patterns reconstructed mental state-specific patterns (mean $Z(r)$ difference $_{matched-mismatched}$ = 0.0016, $d$ = 0.50, $p_{corrected}$ = 0.039). However, we did not find statistically significant evidence that summed mental state patterns could reconstruct situation patterns (mean $Z(r)$ difference $_{matched-mismatched}$ = 0.00016, $d$ = 0.078, $p_{corrected}$ = 0.97). These results indicate that both situations and mental state representations may be composed, at least in part, by sums of the actions they afford. In contrast, mental state affordances do not seem to play a role in the way the brain represents situations.

In addition to these preregistered confirmatory analyses, we also preregistered a matching set of pattern summation analyses reversing the direction of the summation (Fig. S5). These exploratory analyses examine whether action representations can be thought of as sums of the situations and mental states in which they occur, or whether mental states representations can be thought of as sums of the situations in which they occur. In these analyses, we found that summed situation patterns do not reconstruct action-specific patterns (mean $Z(r)$ difference $_{matched-mismatched}$ = 0.0011, $d$ = 0.29, $p_{corrected}$ = 0.37) nor mental state-specific patterns (mean $Z(r)$ difference $_{matched-mismatched}$ = −0.00011, $d$ = −0.081, $p_{corrected}$ = 1.00). However, summed mental state patterns did significantly reconstruct action-specific patterns (mean $Z(r)$ difference $_{matched-mismatched}$ = 0.0018, $d$ = 0.50, $p_{corrected}$ = 0.033). This suggests that the way the brain represents actions can be explained, at least in part, in terms of the mental states that potentiate those actions.

The preregistered pattern summation analyses indicate that affordance-weighted sums of action representations can reconstruct situation and mental state representations. However, they do not indicate what proportion of actions contribute to each situation or state. In principle, it is possible that only a single action (e.g., the most likely one) systematically resembles each situation/state, with the other actions contributing only noise. To rule out this possibility, we conducted three additional exploratory analyses.

First, we directly compared how well affordance weighted sums of action patterns, versus the single most likely action, reconstructed situation and mental state patterns. Reconstruction accuracy was measured using root mean square error (RMSE). For both situations (mean RMSE difference = 0.38, $d$ = 7.45, $p$ = 2.09 × 10−25) and mental states (mean RMSE difference = 0.38, $d$ = 11.00, $p$ = 6.38 × 10−30) we

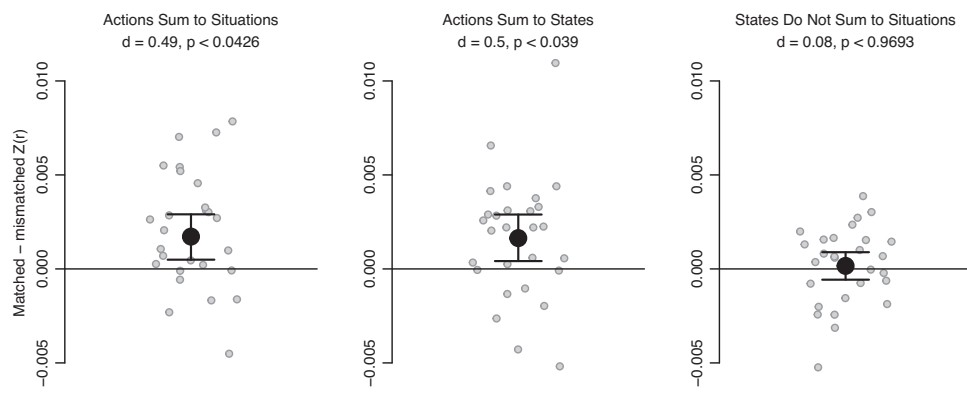

**Fig. 4 | Preregistered confirmatory pattern summation analyses corroborated our hypotheses that frequency-weighted sums of action patterns could reconstruct situation patterns and mental state patterns.** Sums of mental state patterns did *not* reconstruct situation patterns. Reported *p*-values were derived from permutation testing on two-tailed, one-sample *t*-tests, with familywise error rate controlled via the maximal statistic (*n* = 28 participants). *Y*-axis values reflect differences in Fisher z-transformed [$Z(r)$] pattern correlations between matched and mis-matched patterns. Error bars represent 95% bootstrapped confidence intervals around the mean (black). Individual subjects are shown in grey. Source data are provided as a Source Data file.

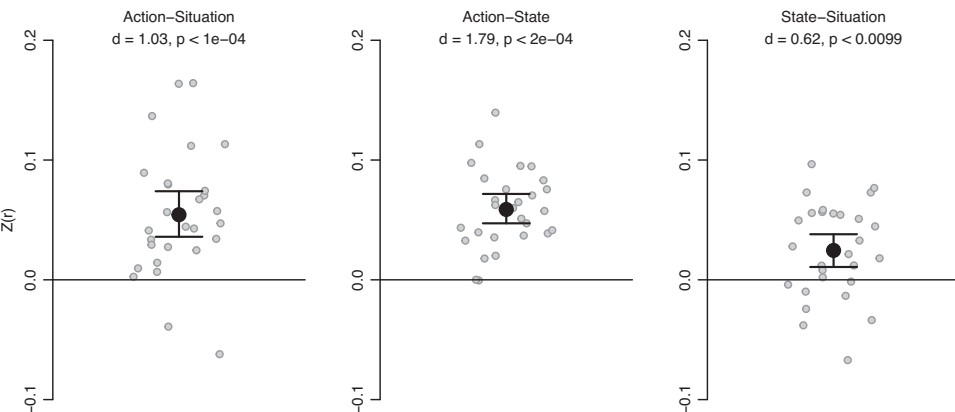

**Fig. 5 | Unregistered representational similarity analyses provided a more general test of our hypothesis that co-occurrence rates shape neural representations of social stimuli.** Neural pattern similarity was significantly correlated with co-occurrence ratings for all three pairwise comparisons between stimulus types. Reported *p*-values were derived from permutation testing on two-tailed, one-sample *t*-tests, with familywise error rate controlled via the maximal statistic (*n* = 28 participants). *Y*-axis values reflect differences in Fisher z-transformed [Z(r)] pattern correlations between matched and mis-matched patterns. Error bars represent 95% bootstrapped confidence intervals around the mean (black). Individual subjects are shown in grey. Source data are provided as a Source Data file.

observed significantly higher RMSEs for the single most likely action than for the weighted sum of actions. This result shows that the affordance-weighted sums of action representations do indeed resemble situation and mental state representations more closely than do the single most likely action for each situation or state.

Second, we compared the summed affordance and single most likely action models by regressing situation and mental state patterns onto one or both of these possible reconstructions. Model performance was measured using the Akaike information criterion (AIC). For both situations (mean AIC = 1482) and mental states (mean AIC = 1700) the single most likely action model achieved the highest AIC (i.e., worst performance). The affordance-weighted sum of actions achieved better performance for both situations (mean AIC = 1480) and states (mean AIC = 1697), and the difference was statistically significant for mental states (mean AIC difference = 2.77, *d* = 0.59, *p* = 0.0041) but not situations (mean AIC difference = 1.92, *d* = 0.36, *p* = 0.066). The model including the weighted sum and single most likely action achieved the best performance for both situations (mean AIC = 1470) and mental states (mean AIC = 1683). This model significantly outperformed both the weighted sum model for situations (mean AIC difference = 10.40, *d* = 1.55, *p* = 8.5 × 10⁻⁹) and states (mean AIC difference = 14.18, *d* = 2.53, *p* = 2.0 × 10⁻¹³), and the single action model for both situations (mean AIC difference = 12.32, *d* = 2.72, *p* = 3.5 × 10⁻¹⁴) and states (mean AIC difference = 16.95, *d* = 2.80, *p* = 1.8 × 10⁻¹⁴). These results further indicate that neural representations of situations and mental states are indeed comprised of sums of representations of the actions they afford. However, the superior performance of the model featuring both the summed affordance and single most likely action patterns also suggests that the highly likely actions receive disproportional weight in the summed representations.

Following up on this last finding, we used representational similarity analysis to non-parametrically estimate the shape of the optimal function for weighting actions by their co-occurrence rates when reconstructing situations and mental states. The results suggest a nonlinear weighting function, such that the most likely actions receive particularly high weights, and the least likely actions receive negative weights (see Supplementary Materials).

The third and final approach we used to rule out the possibility that the single most likely action (or indeed, any single action) could explain neural representations of situations and mental states consisted of a regression-based variant of the pattern summation analyses using cross-validated model selection (see Supplementary Materials). This procedure allowed us to estimate how many actions contributed to each situation and mental state representations. Results indicated

that approximately 44% of actions contribute to optimally reconstructing each situation pattern, and 40% of actions contribute to optimally reconstructing each mental state pattern. Moreover, the regression weights of these optimal decompositions of each situation/state were significantly correlated with the corresponding action affordances. Together, these results thus further reinforce the conclusion that the brain represents situations and mental states as weighted sums of the actions they afford.

To complement the pattern summation analyses, we conducted a set of exploratory representational similarity analyses. Compared with the pattern summation analyses, these representational similarity analyses offer a more general test of our hypothesis that social stimulus representations are shaped by affordances, without the strict requirement that representations of stimuli of one type sum up to representations of another type. In the first set of these analyses, we created three neural pattern similarity matrices by correlating all action patterns with all situation patterns, actions with mental states, and situations with mental states, within the areas of overlap between each pair of stimulus types. Note that, unlike most similarity matrices used in representational similarity analyses, these matrices were asymmetric, because the rows and columns corresponded to stimuli from different domains (e.g., situations and actions). We then correlated these similarity matrices with corresponding ratings of the likelihood of co-occurrence between these stimuli. The resulting participant-level correlations were then entered into one-sample t-tests across participants for inferential purposes. We found that co-occurrence ratings predicted neural pattern similarity in all three cases (Fig. 5), including between actions and situations (mean *Z(r)* = 0.054, *d* = 1.03, *p*corrected = 0.00020), between actions and mental states (mean *Z(r)* = 0.059, *d* = 1.79, *p*corrected = 0.00010), and between situations and mental states (mean *Z(r)* = 0.025, *d* = 0.62, *p*corrected = 0.0099). Performing the same analyses using Kendall's τ yielded qualitatively identical results (see Supplementary Materials). The results of the representational similarity analyses further corroborate the hypothesis that co-occurrences between social stimuli shape how the brain represents them.

We also conducted another representational similarity analysis to test whether action affordances shape situation and mental state representations across the full set of voxels implicated in these two respective domains of social stimuli, rather than just the voxels which overlap with action representation. We found that the similarity between situation representations, but not between mental state representations, was significantly predicted by how similar their action affordances were (see Supplementary Materials).

## Discussion

What are the building blocks of the social mind? This investigation suggests that the brain represents the exogenous and endogenous movers of the social world – situations and mental states, respectively – as sums of the observable actions they afford. That is, the building blocks of one's understanding of a situation comprise – at least in part – the actions that a person is likely to perform in that situation. Likewise, the building blocks of one's understanding of another's mental state comprise the actions that states tend to potentiate. These findings shed light on questions in social, affective, personality, and ecological psychology.

Rich psychological traditions have investigated both the external and internal drivers of behavior. Social psychology has historically focused more on how the external environment – a person's situation – influences their behavior[6,13,14]. Personality psychology and affective science have investigated the internal, endogenous that determine a person's behavior – their enduring personality and momentary mental states, respectively[2,5]. The present findings reveal a link between these related traditions, thereby bridging the gap between external and internal influences on behavior. Specifically, the results suggest that people's understanding of both the internal and external influences is constructed from the same building material: an understanding of other people's actions. People build up useful predictive representations of situations by attending to which actions occur in which external circumstances. Likewise, by attending to which actions people tend to perform when they are engaged in particular modes of thought or feeling, people can form useful predictive representations of others' mental states. Thus, representations of both situations and mental states are constructed, at least in part, from representations of the actions they afford.

Here we observed that action representations form the building blocks of mental state representations. However, mental state representations may, in turn, form the building blocks of even more complex social constructs. Prior research using a similar approach to the one we used here suggests that mental states form the basis of person perception. Specifically, using pattern summation analyses, we previously found that mental state representations can be summed up to reconstruct the neural representations of specific people[30]. That is, the brain represents people as sums of the mental states they habitually experience, just as it represents mental states as sums of the actions those states afford. Considering these results together suggests a hierarchical structure to social knowledge, in which representations of actions are used to construct representations of mental states, which are, in turn, used to construct representations of people. This approach may allow the social mind to efficiently build up useful predictive representations of multiple layers of social knowledge[9].

It is likely that other social dynamics beyond action affordances also contribute to shaping representations of social stimuli. For example, recent research suggests that transition probabilities between mental states causally influence mental state representation[29]. Increasing the likelihood that one mental state follows another causes a corresponding increase in conceptual similarity between these states, and leads people to judge them as closer together on dimensions such as valence. Summed action affordances and inter-state dynamics may thus both contribute to the formation of mental state representations. It is possible that considering all such social dynamics together may provide a full accounting of the structure of social knowledge.

In future work, it will also be important to consider how different types of information may work together to refine people's predictions. For example, knowing someone's mental state alone may help one make broad predictions about what actions they are likely to engage in, but is rarely enough to specify exactly which action they will perform. However, if one has additional knowledge – such as what situation they are in – this could allow one to make much more specific predictions, over and above what the mental state and situation predict cumulatively. The interaction between predictors of behavior could even reverse the direction of certain predictions. For example, if you know someone was at a party, you might make opposite predictions about their mental state based on a second piece of information, such as whether they are more introverted or extraverted. Understanding the general principles behind such interactive predictions is an important challenge for social cognition research.

Recent years have witnessed the introduction of new, well-validated taxonomies of both situations and mental states[35–37]. The present finding may provide a unified *raison d'etre* for the dimensions of situational and mental state taxonomies – that is, why researchers have discovered that certain dimensions describe these domains, and others do not. Specifically, our results suggest that the dimensions of these taxonomies may describe patterns of action affordances. For example, the existence of a valence dimension in both situation[36,37] and mental state taxonomies[35,38] likely reflects the fact that positive situations/states afford very different actions than negative situations/states. Certain actions, such as laughing or dancing, may occur much more often in some situations/states (i.e., positive) than other situations/states (i.e., negative), leading to the emergence of a valence dimension at the situation/state level when such actions are summed to create situation and state representations.

The present results also shed light on important questions in affective science. Constructionist accounts of emotion have become more influential in recent years[39], but the precise computational mechanism by which emotion concepts are constructed is still being established[40]. The present findings lend more weight to accounts that argue that integrating information over time is an essential component of emotion construction[41]. Observable actions are noisy indicators of underlying mental states: one does not *always* get into a fight when one feels angry, even if anger makes this behavior more likely. However, by observing many actions and other indicators of mental states over time, people can build up an understanding of the regularities connecting actions and emotions. This, in turn, can allow them to more effectively use actions to infer emotions, or, conversely, use knowledge of another person's mental state to predict their behavior. Similarly, by observing which actions tend to systematically occur in the presence of which environmental features, people can learn about the types of situations they are likely to experience in their culture, judge the similarity between situations to support better generalization, and predict the behaviors which others are likely to perform in the future.

In exploratory analyses (see Supplementary materials), we examined precisely how, and how many, action representations sum up to situation and mental state representations. Following previous work, the assumption of our preregistered hypotheses was that situation/state representations would reflect a weighted average of the actions they afford[30]. Under this account, unlikely actions would receive weights near zero, and more likely actions would receive increasing weights in linear proportion to their co-occurrence probabilities. However, this is not the only weighting function which might make sense. Indeed, results of our supplementary analyses hint at two ways in which the optimal weights might differ (Fig. S6). First, the weighting function is not linear: highly likely actions contribute disproportionately more to situation and state representations. Second, weights can be negative. This suggests that situations and states are defined not only by the presence of actions they do afford, but also by the absence of actions they don't afford. Additionally, cross-validated model selection results provide further evidence that multiple actions – 40–44% in this sample – contribute to optimally reconstruct each situation/mental state, and that the magnitudes of those contributions are correlated with affordance ratings, further corroborating the general hypotheses of this investigation.

The observation that mental states are defined, at least in part, by the distribution of actions they potentiate suggests a connection with

the class of statistical models known as Hidden Markov Models (HMMs). These models consist of a set of unobservable latent states, characterized by two features: (i) a set of transition probabilities between states, and (ii) a set of probabilities specifying how often each state "emits" various observable indicators. In the social-affective case, mental states/emotions would play the role of latent states, and people's actions would play the role of emissions. We have previously shown that there are highly reliable transition probabilities between different emotions, and that people have accurate knowledge of these probabilities[42]. Combining this result with the present findings suggests that HMMs may provide highly effective computational models for how the brain conceptualizes others' actions, affect, and the relations between them.

This investigation also provides support for a longstanding hypothesis in ecological psychology: that situations are composed of the actions they afford[3,7]. Despite the representational perspective we have adopted here, our findings nonetheless bolster prior arguments that action affordances play a key role in perceiving, predicting, and reacting to the world around us[1,12,19,43]. They also support prior findings indicating that affordances are not only relevant to understanding actions and situations, but also to understanding other people's emotions and dispositions[19,44].

More generally, our findings are also consistent with theories of predictive coding, which suggest that neural representations of the external world consist primarily of active predictions about the future state of the world, rather than passive perceptions of the world as it currently is[8,22]. Specifically, our findings suggest that people's understanding of situations and mental states resemble bundles of predictions about the actions likely to occur in those situations/states. In Bayesian terminology, situations and mental states represent prior probability distributions on actions, conditioned upon either external or internal variables. One key prediction this perspective makes is that the real-world co-occurrence rates between situations/states and actions should be correlated with the perceived co-occurrence rates we examined here. Future research could test this hypothesis using methods such as experience sampling. Based on prior work, we expect that people's perceptions of action affordances will be largely accurate, but may feature certain well-documented distortions in statistical thinking, such as base-rate neglect[42,45].

Although the results of this study generally corroborated our primary hypotheses, one exception bears further discussion. Specifically, the data indicated that mental state affordances do *not* sum up to situation representations. We initially hypothesized this relation because certain mental states do seem more or less appropriate in different situations. For example, it would be distasteful, or downright suspicious, to express glee at a funeral. Our participants agreed (Fig. S3), assigning considerably different ratings to different state-situation co-occurrences. However, unlike in the cases of situation-action and state-action co-occurrences, we did not observe that mental states summed to the situations with which they frequently co-occur. Despite this, we did observe a significant correlation between neural pattern similarity and situation-state co-occurrences in our exploratory representational similarity analyses. Together, these results suggest that situation and mental state representations are indeed related within the same brain regions, but not via a process of summation of one into the other. Future research should investigate alternative explanations of this relationship.

The present investigation marshals evidence for its conclusions by adopting several important practices. First, the study was preregistered in detail, including an a priori power analysis, the behavioral task specifications, and the analytic approaches. Despite some deviations described herein, and the addition of several exploratory analyses, the investigation generally hewed closely to registered plans. This improves transparency and constrains analytic flexibility, important considerations in an investigation of this complexity. Second, we optimized our

stimuli via a two-step process using both computational text analysis and human judgements. Moreover, the partially-crossed design used in the imaging studies maximized the variability in the combinations of these stimuli that participants judged in the scanner. These choices improved both our statistical power and our ability to generalize from the specific stimuli we selected to the broader psychological domains of actions, mental states, and situations from which we sampled[46]. Third, we adopted a combination of data-driven methods to identify brain regions representing situations, mental states, and actions, and theory-driven hypotheses, to guide our confirmatory analyses. This combination of data-driven and theory-driven approaches offers the benefits of both approaches by allowing the data to speak for itself, yet constraining it to say something meaningful. Finally, the statistical analyses we applied were highly conservative in multiple respects, including our approach to controlling for multiple comparisons, and our use of data from separate trials for each domain in the pattern summation analyses (e.g., we summed up action patterns measured when participants were not making judgements about situations to reconstruct situation patterns which were measured when participants were not making judgements about actions). These choices set a high bar for corroborating the hypotheses we tested.

Nonetheless, this investigation does have important limitations which we ought to highlight. First, our samples were not representative of the US or global population, having been convenience-sampled from US college students and Mechanical Turk workers. Moreover, due to the Covid-19 pandemic, our imaging sample was smaller than we planned, albeit still offering a high level of statistical power according to our a priori power analyses. These limitations on our samples constrain the generalizability of the present results. Second, we used only one task to assess participants' neural representations of situations, mental states, and actions, and that task was not naturalistic. Corroborating our findings using a broader array of more naturalistic tasks would improve the convergent validity and external validity of the conclusions. Finally, although we report statistically significant results with moderate-to-large standardized effect sizes, the raw effect sizes indicate that summed action affordances are far from perfect reconstructions of situation or mental state affordances. Moreover, situation, mental state, and action representations only partially overlap in the brain. This means that, even if we had observed very high reconstruction accuracies, the summed affordances could not be a complete account of the formation of these social representations (although supplementary analyses suggest that action affordances describe situation representations beyond regions of action representing voxels). Thus, while the results indicate that action affordances play a role in constructing situation and mental state representations, other factors must also contribute.

This investigation sought to understand how people understand the external situations and internal mental states that predict others' actions. The results suggest that representations of situations and mental states are composed – at least in part – of probability-weighted sums of representations of the action those situations and states afford. This outcome indicates that people's understanding of the social world's exogenous and endogenous forces may be constructed from the same set of building blocks: actions. The results also lend weight to the hypothesis that prediction is the fundamental goal around which social cognition is organized[26]. Taken together with other recent studies which have adopted a similar perspective[30], this work corroborates the model of the social mind that encompasses multiple layers of social knowledge, all mutually predicting each other and themselves[9].

## Methods

This research complies with all relevant ethical guidelines, and was approved by the Princeton University Institutional Review Board (protocol #0000007271). We report how we determined sample sizes,

all data exclusions, all manipulations, all measures, and all deviations from our registered plans. All statistical tests were two-sided. Data and code from this study and all others in this paper are freely available on the Open Science Framework (OSF; https://osf.io/qwd2k/). Neuroimaging data are available on OpenNeuro.org (https://openneuro.org/datasets/ds004226/). This study was preregistered on OSF (https://osf.io/whsx8).

## Participants

A parametric power analysis was conducted to determine an appropriate sample size for the imaging portion of this investigation. To this end, we drew on a previous paper[30] where we tested a similar hypothesis (i.e., whether neural representations of people could be reconstructed by summing the representations of mental states they frequently experienced). That paper featured two studies with similar designs and analyses those here. To be conservative, we targeted the smaller of the two resulting effect sizes: Cohen's $d = 0.81$. A one-sample $t$-test power analysis was conducted using the 'pwr' package in R[47]. This indicated that a sample size of 34 would be necessary to achieve 95% power, correcting for the effect of multiple comparisons (i.e., across our three primary hypothesis tests described below) on both type I and type II error rates.

Our preregistered inclusion criteria required participants to be right-handed (or ambidextrous), 18–35 years of age, fluent in English, have normal or corrected-to-normal vision, have no neurological abnormalities, and have no safety contraindications for the MRI environment. These criteria were adhered to, except for one participant who was 36 at the time of scanning. Imaging participants were paid $20 per hour. Using SONA systems for recruitment, we had completed data collection from 29 participants when the outbreak of the COVID-19 pandemic led to a shutdown of our imaging center. During the period that the imaging center was shut down, key study personnel dispersed to different institutions, creating a barrier to further scanning. As a result, we did not complete collection of our original targeted sample size. To maximize the amount of data that could be used, we relaxed our preregistered exclusion criteria. The revised exclusion criteria were a subset of the original criteria. Specifically, we excluded participants if, for five or more runs, any of the following applied: they responded to fewer than 70% of trials, moved more than 2 mm, and/or moved more than 0.5 mm 5 times. This led to the exclusion of one participant, due to head motion. The remaining sample comprised 28 participants (16 female, 12 male; mean age = 20.61; age range = 18–36; 13 Asian, 9 White, 2 Black, 3 multiracial, 1 other). Sex was self-reported, and was not a component of study design. Sex and gender analyses were not conducted as we did not have hypotheses regarding these variables, and we did not collect a sufficient sample size to analyze sex and gender categories separately or compare them. Results cannot be disaggregated by sex or gender due to the use of multi-individual composites in stimulus selection, feature selection, and independent variable construction. Despite not achieving our intended sample size, our sample still offered statistical power in excess of the typical gold standard (80%), with a family-wise power of 86%, and 98% power for each individual hypothesis, based on our original a priori power calculation.

In addition to our primary sample of imaging participants, we collected two additional online samples. One of these samples rated situations on the dimensions of the DIAMONDS taxonomy (total $N = 400$; 7 excluded for reporting less than fluent knowledge of English; 177 female, 209 male, 4 other, 3 prefer not to state; mean age = 37.01; age range = 19–81). These ratings were used to facilitate stimulus selection, as described in the next section. The other online sample rated the probability of co-occurrences between situations, mental states, and actions ($N = 900$; 290 female, 580 male, 30 declined to state; mean age = 25.99; age range = 18–66). These ratings were combined with the neuroimaging data to perform our primary hypothesis

tests. Both samples were recruited from Amazon Mechanical Turk via CloudResearch[48] and paid $6.50/hr. The sample sizes for these studies were based on our prior experience norming stimuli in studies of similar design, with the aim of generating reliable composite averages across participants. All imaging and online participants provided informed consent in a manner approved by the Princeton University Institutional Review Board.

In addition to these data, we used publicly available data from two prior investigations of mental states and actions, respectively[49,50]. These data consisted of ratings of mental states and actions on the dimensions of the 3d Mind Model and ACT-FAST taxonomy. These ratings were used to facilitate stimulus selection, as described in the next section.

## Stimuli

This investigation presented participants with three classes of stimuli: situations, mental states, and actions. All three classes were presented verbally by individual words representing mental states (e.g., "relief" or "jealousy") and actions (e.g., "eating" or "dancing") or by short phrases representing situations (e.g., "at the gym" or "with an annoying acquaintance"). An initial set of 472 actions was generated based on verbs studied in previous research[50]. This set was manually reduced to 224 by eliminating actions we judged too polysemous when presented as a single word. An initial set of 166 mental states was generated based on a prior study[49]. An initial set of 166 situations was generated manually by the research team. These descriptions were designed to be similar in length, and to avoid explicitly describing the specific mental state or action of the person in the situation. Instead, they focused on physical context (e.g., time and place) and/or social context (e.g., other people present).

After generating the initial stimulus sets, we performed two rounds of stimulus optimization to create the final sets for use in the fMRI experiment (Fig. 1). The first round of stimulus optimization relied on computational text analysis. For each mental state and action word, and for each word within each situation vignette, we extracted a corresponding word vector from the fastText embedding[51]. These 300d vectors represent the meaning of words, measured via statistical regularities in large bodies of text. Vectors for the situations were averaged across words. Several stimuli did not produce unique vectors, leaving us with 220 actions, 163 mental states, and 166 situations. We assessed the similarity in meaning between stimuli in different classes by correlating their vectors (i.e., correlating situation vectors with state vectors, state vectors with action vectors, and action vectors with situation vectors). These correlation matrices became the basis for this round of stimulus selection.

This optimization procedure aimed to select subsets of 100 stimuli from each class, which maximized the variance in their similarity to stimuli in the other classes. We set this goal based on the principle that word vectors are heavily influenced by word co-occurrences, and that word co-occurrences serve as a proxy, albeit imperfect, for real-world co-occurrence[52]. By selecting stimuli that varied in their word vector similarity to each other, we hoped to thereby select situations, mental states, and actions that maximally varied in their real-world co-occurrences. To achieve this high-level goal, we simulated 10,000 random draws of 60 stimuli from each class (60 being our final target number of stimuli from each class). We then computed the standard deviation of the similarities between each stimulus and the stimuli from the other classes. For example, if "applauding" was one of the actions in the current set of 60, we would compute the standard deviation of the similarities (i.e., correlations) between applauding and each of the mental state words, and between applauding and each of the situations. We recorded these standard deviations for all stimuli across all 10,000 random draws, and then averaged them across draws. After z-scoring, we also averaged across the two different classes that each class was paired with (i.e., for actions, the mean SD

across mental states and situations). This produced an index for each stimulus that represented how well it satisfied our variance-maximizing goal. We selected the 100 stimuli in each class that achieved the highest scores on this index.

The second round of stimulus optimization was based on human ratings of the stimuli. From prior investigations[49,50], we already had access to ratings of the actions on the dimensions of the ACT-FAS Taxonomy (abstraction, creation, tradition, food, animacy, and spiritualism) and ratings of mental states on the dimensions of the 3d Mind Model (valence, social impact, and rationality). Since the situation stimuli were newly created for this study, we collected ratings of the situations on the dimensions of the DIAMONDS taxonomy (duty, intellect, adversity, mating, positivity, negativity, deception, and sociality) from a new online sample[37]. Participants in this sample rated all 100 situation stimuli on just one of the DIAMONDS dimensions, using a 7-point Likert-type scale. These ratings were averaged across participants to create composites for use in stimulus selection.

The first round of stimulus optimization aimed to maximize variance in stimulus co-occurrences. The goal of this second round was instead to minimize redundancy. This included minimizing redundancy between the stimuli themselves and minimizing redundancy between the psychological dimensions that described these stimuli. To achieve these goals, we used a greedy search procedure. This procedure began by randomly selecting 60/100 stimuli within a given class (situations, states, and actions were optimized independently in this round). Then, on each of 10,000 iterations, a potential substitution of one of the 60 stimuli for one of the left-out 40 was considered. This substation was evaluated on two criteria. First, we estimated the redundancy between stimuli by computing the Euclidean distances between them within the respective psychological rating space (e.g., ACT-FAST for actions). To do so, we computed the distance between each stimulus and each other stimulus within the set of 60 that would be generated if the substitution took place. We then averaged these minima across the 60 stimuli. Second, we computed the correlations between the rated psychological dimensions across the stimuli. We computed summed the (negative) maximum and mean absolute correlations between these dimensions as a measure of redundancy (i.e., maximizing orthogonality). We then summed these two criteria to produce a final measure of how well the substitution under consideration would achieve our goal of redundancy minimization. If the substitution met that goal better than the current set of 60, then that substitution would be performed. If not, the current set of 60 would be retained. In either case, another random substitution would be considered on the next iteration.

In sum, these stimulus selection procedures allowed us to generate sets of 60 situations, mental states, and actions which maximized variance in co-occurrence rates between these classes and minimized redundancy of stimuli and psychological dimensions within these classes. These final sets of stimuli were rated by a separate sample of online participants, who judged the likelihood of co-occurrences between them (e.g., how likely is it that a certain action would occur in a certain situation?). Participants made these judgements on 100-point line scales. We averaged across participants (an average of 10 ratings per stimulus pair) to generate composites that would be used in our primary imaging hypothesis tests. These same sets of stimuli were also viewed and judged by participants in the fMRI scanner, as described below.

## FMRI paradigm

Participants undergoing fMRI scanning made judgements about the likelihood of co-occurrences between situations, mental states, and actions. On each trial, they would be presented with two stimuli from different classes (i.e., a situation and a state, a situation and action, or a state and an action). For example, on a situation-action trial, they might be asked to judge "How likely is it that a person in this situation:

at a funeral, is engaging in this action: dancing?" Participants responded on a 1–4 scale anchored at "not at all likely" and "very likely" using a button box in their left hand. Each trial lasted 4250 ms and was followed by a minimum fixation period of 250 ms, plus an additional jittered fixation period. The jitter was Poisson distributed in 1.5 s increments, with a mean of 1.5 s. This task was programmed in Python[53] using PsychoPy[54].

Participants completed 10 runs of this task over the course of the experiment. Each run consisted of 90 trials, including 30 situation-state pairs, 30 situation-action pairs, and 30 state-action pairs. Within each run, participants would see only 30 situations, 30 states, and 30 actions (twice each). This meant that the same situations, states, and actions were used evenly for each type of pairing (i.e., the 30 states paired with situations on a given run would also be the 30 states paired with actions on that run). This allowed us to strictly control how often each stimulus co-occurred with the two other classes (i.e., so that a certain action didn't occur more with situations than with mental states). Across pairs of runs (e.g., runs 1 and 2), the full set of 60 stimuli of each class would be presented. The sets of 60 were randomly divided in half independently for each pair of runs (e.g., the run 1–2 split would be different from the run 3–4 split) and across participants and stimulus classes. Thus, over the course of the 10 runs, each stimulus was guaranteed to be presented exactly 10 times. However, the particular pairings between stimuli of different classes were randomized within each run, subject to the constraint that no particular pairing was repeated within-subject. This produces a partially-crossed design at the level of pairings (i.e., there are $60 \times 60 \times 3 = 10{,}800$ unique pairings between stimuli, but each imaging participant saw only 900 of them). This design helped to maximize the variety of stimulus pairing across participants, and thereby the generalizability of our findings. A short practice version of the task was presented to participants before fMRI data collection to familiarize them with the procedure. This practice version used stimuli that were not used in the main task.

In addition to the fMRI task, participants completed a series of post-scan measures, including a 2-item extraversion measure, the Interpersonal Reactivity Index[55], the Narcissistic Personality Inventory[56], Autism Quotient[57], and the revised Reading the Mind in the Eyes task[58]. Participants also rated the transition probabilities between a set of mental states and provided their demographic information. These measures were collected to facilitate potential individual difference analyses in combination with other imaging datasets collected by the lab and will not be discussed further in this paper.

## Imaging procedure

All imaging data were collected on Siemens Skyra 3 Tesla scanner (Siemens, Erlangen, Germany) with a 64-channel head coil. Functional echo-planar BOLD images were collected with TR of 1500 ms; TE of 32 ms; flip angle of 70°, and spatial resolution of 2.5 mm isotropic voxels. The preregistration called for 2 mm voxels, but after piloting, we decided that a larger voxel size produced better signal. Slices (52) were acquired in an interleaved, axial fashion with a simultaneously multislice acquisition factor of four. In addition to BOLD EPIs, we acquired a high-resolution anatomical image from each participant for the purposes of intersubject alignment. These images were generated by a T1-weighted scan with 1 mm isotropic voxels, a TR of 2300 ms, TE of 2.98 ms, flip angle of 9°, and 176 slices. We also collected two spin echo field maps (phase encoding A » P and P » A) to correct for inhomogeneities in the magnetic field via unwarping. These scans featured 2.5 mm isotropic voxels at a TR of 8000 ms and TE of 66 ms, with 52 transversal slices. A localizer and AA scout were used to determine the position of participants' brains and align scans accordingly.

After data collection, imaging data were subjected to preprocessing and general linear modeling (GLM) using the multipackage Data Analysis Modules for Neuroimaging pipeline (https://github.com/PrincetonUniversity/prsonpipe). Using this pipeline, we applied

SPM12 for slice time correction[59], DARTEL for head motion correction, unwarping, and normalization[60], and FSL for high pass filtering[61].

After primary preprocessing, a GLM was applied to generate patterns of brain activity associated with each situation, mental state, and action. The preregistration called for a single GLM to perform this estimation, but this was based on a separate set of hypotheses distinct from the investigation reported here. That set of hypotheses specified the use of Bayesian representational similarity analysis. Bayesian models are less restricted than ordinary least squares versions of the same analysis, in that they can estimate posteriors over linearly dependent effects. However, that is not the case for the GLM we intended to use here, creating a collinearity issue. Specifically, each of the trials would have been modeled twice. For example, a situation-action trial would have been modeled once as a situation, and once as an action. This would have made the regressor matrix rank deficient: any of the regressors could have been predicted from a linear combination of the others, making the regression parameters impossible to estimate. To address this issue, we instead estimated activity patterns using two separate GLMs. One GLM modeled situations in the context of situation-action trials, situations in the context of situation-state trials, and states in the context of state-action trials. The other GLM modeled states in the context of situation-state trials, actions in the context of situation-action trials, and actions in the context of situation-state trials. GLMs were implemented in MATLAB (https://www.mathworks.com/products/matlab.html) using SPM12[59] and SPM12w (https://github.com/wagner-lab/spm12w).

Thus, across the two GLMs, there were 360 conditions of interest: 60 situations x 2 (paired with states vs. paired with actions), 60 states x 2 (paired with situations or paired with actions), and 60 actions x 2 (paired with situations or paired with states). Boxcar regressors were created based on the onsets and durations of trials in each condition. Durations were set to reaction times, or the full presentation time if no response was made. The boxcars were then convolved with a canonical hemodynamic response function before being entered into the design matrix. The GLMs also include nuisance regressors for run means and linear trends, and 6 degree of freedom head motion estimates. Each condition was contrasted against baseline, to yield 360 wholebrain activity maps. These maps were then subjected to statistical analysis, as described below.

## Statistical analyses

**Feature selection**. To determine whether neural representations of situations, mental states, or actions sum up to one another, it is first necessary to determine where such representations are located, and more specifically, where they overlap. An absence of overlap would indicate that these different domains of social representation cannot possibly sum up to one another. Although the representations could be related in other ways, a lack of overlapping patterns would make our primary registered hypotheses non-starters. To identify voxels contributing to the representation of situations, mental states, and actions, we used reliability-based feature selection. This procedure allows for the identification of the voxelwise reliability threshold that maximizes pattern-wise reliability, thereby selecting for both highly reliable response properties within voxels and across voxels.

Our preregistration specified no spatial smoothing, but this specification neglected the need to perform smoothing to facilitate the reliability-based feature selection. Since this feature selection entailed averaging patterns of brain activity across participants (unlike our other analyses, which were conducted entirely within subject), spatial smoothing was called for to improve the alignment between participants. Thus, for feature selection only, SPM 12 was used to apply 6 mm Full-Width at Half-Maximum Gaussian spatial smoothing[59]. This was applied to GLM regression coefficients, not the preprocessed BOLD data.

We performed reliability-based feature selection separately for each class of stimuli using custom MATLAB code. Reliability-based feature selection is typically done across the full set of stimuli/conditions in a condition-rich design. However, in most condition-rich designs, all stimuli/conditions are presumed to belong to the same psychological domain and be represented by the same neural substrates. This assumption is not justified in the present case, since we are explicitly investigating three different classes of stimuli: situation, mental states, and actions. Although we hypothesize that they will overlap, it would be inappropriate to assume it without putting this assumption to the test. If we performed reliability-based feature selection across all conditions, we would inevitably detect some set of voxels due to the logic of the procedure. That set of selected voxels would reflect regions that reliably represent situations or mental states or actions, but there would be no guarantee that any of them represents all three (or even two) of these types of stimuli. Thus, by performing feature selection separately for each class, we put ourselves in a better position to critically test an underlying assumption of our main hypotheses. By making it possible to find that there is no overlap between situation, mental state, and action representations, we make our overall hypotheses that much more falsifiable.

In the interests of clarity, we will describe it here only in the case of situations, with the understanding that the same procedure was carried out with respect to mental states and actions. For each voxel in each participant's brain, the GLMs yielded 120 values which indicate how active that voxel was in response to the 60 situations: 60 values corresponding to situations coming from situation-action trials, and 60 values corresponding to situations coming from situation-state trials. We averaged these respective sets of values across participants. We then correlated the two independent sets of 60 values with each other. This produced a wholebrain voxelwise reliability map for situation-related activity.

We set thresholds for this voxelwise reliability map ranging from 0 to the maximum observed voxelwise reliability for each map, in increments of 0.01. At each threshold, we extracted the activity pattern corresponding to each situation. Again, there were 120 such patterns for each participant, corresponding to 60 situation patterns from situation-state trials and the same 60 situations patterns from situation-action trials. These patterns were averaged across participants. We then compute the similarity between patterns within each set of 60 by correlating them with each other. The lower triangular elements of these pattern similarity matrices were then correlated with each other. The resulting value provides an estimate of the pattern similarity reliability at a given voxelwise reliability threshold.

As the voxelwise reliability threshold increases, pattern similarity reliability tends to first increase – as irrelevant voxels are discarded – and then decrease – as important voxels are discarded. We selected an optimal voxelwise threshold using the rise-and-fall dynamic. Specifically, we selected the last voxelwise threshold for which the pattern similarity reliability continued to increase consistently (i.e., the last voxelwise threshold before the first observed decrease in pattern similarity reliability). Voxels with reliabilities above this threshold were included in the subsequent pattern summation and pattern similarity analyses.

It is important note that, like other feature selection techniques, reliability-based feature selection has the potential to produce circularity, if it is inappropriately applied[32,62]. In particular, if the units of analysis over which one calculates reliability are also the only independent or dependent variable in one's analysis, then this carries the risk for circularity. For example, if we were to try to perform a 60-way classification on the situations (i.e., where the situations are the labels), based on the corresponding patterns of brain activity, using reliability-based feature selection would positively bias performance. In that case, it would be necessary to perform feature selection using separate fMRI data from the classification. However, in the present case, the

results hinge on a third variable – co-occurrence ratings – which were not involved in the feature selection process. As a result, feature selection and subsequent analyses can be perform on the same data without bias[63].

**Pattern summation analysis.** Inferential statistics were performed using R 4.0.3[64] including the glmnet[65], DescTools[66], and pracma[67] packages. As predicted, we observed regions of overlap between regions identified by reliability-based feature selection for situations, mental states, and actions. We thus aimed to test whether summing up patterns representing stimuli of one type (e.g., actions) could reconstruct patterns representing another type of stimuli (e.g., situations). To this end, we extracted patterns of brain activity from the GLM results from these overlapping regions. So, for example, from the voxels shared by situations and mental states, we extracted patterns corresponding to each situation, and each mental state. Importantly, these patterns were not based on all of the trials featuring a given situation, state, or action. Rather, for a given set of overlapping voxels, we extracted patterns from different sets of trials. For instance, from within the situation-state overlap, we extracted situation patterns derived from trials on which participants judged situation-action co-occurrences, and mental state patterns derived from trials on which participants judged mental state-action co-occurrences. We did this to avoid biasing the results in favor of our hypotheses. For example, it would be less remarkable if situation patterns were related to action patterns on trials when participants explicitly judged how often situations and actions co-occur, compared to trials on which they judged situation-state or state-action co-occurrences. Thus, we took the latter approach to be more conservative – if anything, biasing the results against the hypotheses we planned to test.

Using all of the stimuli of one type (e.g., actions), we aimed to reconstruct each stimulus of another type (e.g., situations) with a different weighted average (Fig. 3). The 60 action patterns, for instance, would be weighted based on how often our online participants rated each to co-occur with a specific situation. Thus weighted, we would take the voxelwise average of the action patterns. This process was repeated to reconstruct all 60 situation patterns using the action patterns. We repeated this process using actions to reconstruct states, and states to reconstruct situations. In preregistered exploratory analyses, we also reversed the directions of each of these summations (e.g., reconstructing action patterns using averaged situation patterns).

We compared the reconstructed patterns with the real patterns via Pearson correlation. Each reconstructed pattern was correlated not only with the one matching real pattern, but also with other 59 real patterns within the targeted domain. We subtracted the average of the mis-matched correlations from the matched correlation to ensure that the resulting values reflect specific reconstructions of the targeted real patterns, rather than a generic reconstruction of all patterns within a domain. All of these differences were then averaged within participant to produce scalar estimates of how well patterns from one domain could reconstruct patterns from another domain. We tested whether the values were statistically significant by entering them into a one-sample $t$-test across participants. A significant outcome would indicate that patterns from one domain could indeed reconstruct patterns from another. We corrected for multiple comparisons across each family of tests (i.e., within the three confirmatory pairings of domains and separately within the three exploratory pairings) by applying maximal statistic permutation testing to the $t$-tests.

In addition to this primary set of pattern summation analyses, we conducted three additional exploratory pattern summation analyses to test whether multiple actions were indeed required to construct situation/state patterns. First, we directly compared how well affordance weighted sums of action patterns, versus the single most likely action, reconstructed situation and mental state patterns. To this end, we $z$-scored all situation, mental state, and action patterns (separately

for each of these three sets) voxelwise within each subject. The resulting standardized patterns reflect what is unique about each stimulus condition, rather than any global task positive pattern. Next, we computed the RMSE between each situation and state pattern, and the respective affordance weighted sum of action patterns. We also computed the RMSE between the situation/state patterns and the pattern for the single most likely action to occur in the corresponding situation/state. We averaged the RMSEs across the 60 situations/states within participant, and then compared the affordance weighted sum RMSEs to the single most likely action RMSEs via paired $t$-tests.

Second, with each situation/state pattern as the dependent variable, we fit three linear regressions: one featuring just the affordance weighted sum of action patterns, one featuring just the pattern of the single most likely action's pattern, and one featuring both of these predictors. For each of these models, we computed AIC as a parsimony-adjusted model performance metric. We averaged these AICs across all situations, or all mental states, within a participant. We then compared the average AICs for the three regressions to each other via paired t-tests across participants.

Finally, we conducted a cross-validated model selection procedure using L1 regularized regression (see Supplementary Materials). This procedure allowed us to estimate how many actions were contributing to optimal reconstructions of situations and mental states.

**Representational similarity analysis.** To complement the pattern summation analyses, we also conducted a set of exploratory representational similarity analyses[68]. These analyses represent a more general version of our primary hypothesis. They test whether the neural similarity between stimuli in different domains is related to co-occurrence rate, but do not require that patterns from one domain sum up to patterns in another. These analyses were based on the same sets of situation, state, and action patterns used in the pattern summation analyses. However, instead of adding one set up to reconstruct the others, we Pearson correlated all patterns from one domain with all patterns from another domain. For example, all situation patterns were correlated with all action patterns to produce a $60 \times 60$ neural pattern similarity matrix comparing these two domains. We then tested whether the similarity between these patterns correlated with ratings of the co-occurrence of the corresponding stimuli. For example, whether the similarities between actions and situations were correlated with how often those actions and situations were thought to co-occur. To this end, we vectorized both the pattern similarity matrices, and the corresponding co-occurrence rating matrices (visualized in Figs. S1, S2, & S3) and Pearson correlated them with one another. This process yielded one correlation value for each participant. We Fisher transformed these correlations, and then entered them into one-sample $t$-tests across participants to determine whether they were statistically significantly different from change ($r = 0$). We controlled for multiple comparisons via maximal statistic permutation testing.

In addition to this primary set of representational similarity analyses, we conducted three additional variants (see Supplementary Materials). First, we repeated the analyses described above using Kendall's τ instead of Pearson correlation coefficients to estimate the relationship between affordance ratings and neural pattern similarity. Second, we conducted a version of the analyses described above using optimized LASSO regression coefficients, instead of zero-order correlations, to estimate the contributions of action patterns to situation and mental state patterns. Third, we conducted within-domain representational similarity analyses (e.g., comparing situation representation to other situation representations, rather than directly to action representations) using the full set of voxels involved in situation and mental state representation, respectively, rather than just regions of overlap with action representation. Finally, we used representational similarity analysis to nonparametrically estimate the optimal shape of the action co-occurrence weighting functions for situations and

mental states (i.e., the curvilinear relationship between affordance ratings and how much each action contributes to a situation or state representation).

## Reporting summary

Further information on research design is available in the Nature Portfolio Reporting Summary linked to this article.

## Data availability

The raw MRI data generated in this study have been deposited in OpenNeuro database (https://openneuro.org/datasets/ds004226/) and are freely available. The behavioral data and derivate neuroimaging data generated in this study have been deposited on the Open Science Framework (https://osf.io/qwd2k/) and are freely available. Source data are provided with this paper.

## Code availability

Computer code used to present the experiments and analyze the data in this investigation has been deposited on the Open Science Framework (https://osf.io/qwd2k/) and is freely available.

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

## Acknowledgements

The authors thank Elyssa Barrick and Mihir Gandhi for their assistance. This work was supported by NIMH grant R01MH114904 to D.I.T.

## Author contributions

M.A.T. and D.I.T. conceptualized the research. M.A.T. conducted the investigation. M.A.T. contributed software and methodology and conducted the formal analyses. M.A.T. visualized the results. M.A.T. prepared the original draft. M.A.T. and D.I.T. contributed to writing the manuscript. D.I.T. supervised the research.

## Competing interests

The authors declare no competing interests.
