## [Peer Review File · Nature Communications]

Neural representations of situations and mental states are composed of sums of representations of the actions they affordReviewer #1 (Remarks to the Author):

Thornton and Tamir studied how the neural representation of states and situations is linked to the actions that they afford/provide. To test this prediction, the authors used a simple but effective experimental paradigm. They asked participants to report the likelihood of an action in different situations and states (and a similar procedure for situation/state). They then went on and tested their primary hypothesis (that affordances shape social representations) by summing up action representation (weighted by their cooccurrence) and correlating it with the other representation (state or situation). The authors then tested the similarity between actions, situations, and states using RSA. In both sets of analyses, the results were consistent with the prediction that the authors made in the introduction.

The research question is very interesting and timely. The study was preregistered which is commendable, and the statistical procedures are solid. However, I do not believe I agree with the main conclusion of the study that weighted sum of actions shapes the representation of the states that they occur, at least as it stands now. In the following I will elaborate more on this:

1)The RSA results nicely indicate that there is a correlation between similarity matrices (between action and states) with corresponding ratings of the likelihood of co-occurrence between these stimuli. This indicates that the degree to which the representation of action A is similar to state S depends on the likelihood that A occurs in S. Therefore, if S affords three actions (A1, A2 and A3), it is potentially correlated with all three actions (though the strength of the correlation varies between actions, obviously depending on the cooccurrence). This means that it is very likely that S is also correlated/similar to A1+A2+A3 (if B is correlated with A1, A2, A3, it is highly likely that it is correlated with their sum too). The authors have done a similar analysis (though not using RSA) and have concluded that S represents the summed actions it affords. Without doing formal model comparison, it is impossible to judge whether the representation of S is only similar to the weighted sum of the action or only one of them (e.g., the most likely one).

There is a recent development in testing different models of representational geometry (see <https://arxiv.org/abs/2112.09200>). I highly recommend the authors to use this tool to test different models (i.e., the weighted sum and individual action models). The toolbox can be used to estimate the weight of different RDMs (the actions afforded by a state or situation in). The authors can test whether the weighted model is the best model and whether the weights estimated by the representational model is correlated with the reported probability of the occurrence. I believe without such a robust model comparison, the authors cannot claim that the representational geometry is “best” explained by the sum of the actions.

2) in many situations, we know someone’s mental state (like anger) but we do not know what they might do when they are in that state. What does the framework suggested by this study predict for these situations?

2)The authors report that “A set of 570 voxels were shared between the situation and action maps, 668 voxels were shared between the mental state and action maps, and 1545 voxels were shared between the situation and mental state maps. The existence of this spatial overlap provides an initial indication that neural representations of situations, states, and actions may be relate”. I could not find any information about the spatial map of these voxels. Can the authors report such a map? It can be useful for understanding the computational role of some of the key brain regions in social cognition. For example, Roumazeilles et al. Science Advance, 2021 shows that an area in mid-STS (an

area most similar to human TPJ) responds to social prediction error. Did the authors find any voxels in this area of the human brain?

Minor:

1) It would be very useful if the authors relocate Figure 5 to the result section where they introduce their methods

Reviewer #2 (Remarks to the Author):

This paper aims to test whether and how individuals represent situations and states as a sum of associated action affordances. The research question, introduction, and theoretical background are well described and clear. The incorporation of the predictive coding theory to predict people's actions in social situations/states is interesting. The stimulus optimization procedure, described in detail, is clear, and the discussion is well-written.

However, I have some major concerns regarding the methodological approaches used, particularly for the fMRI data.

First, the authors are performing voxel feature selection based on the reliability-based feature selection method. The critical advantage of the reliability-based voxel selection method (Tarhan & Konkle) is that experimenters can select reliable voxels independent of main condition effects and data. For instance, when selecting voxels for object size analysis (comparing big vs. small objects), all object trials (e.g., 100 objects) should be considered, regardless of object size. Voxels with high reliability (within > across activity for each trial type) can then be selected. This allows the selection of voxels independent of the question of interest, which is about the size of objects.

The problem with the voxel feature selection approach used in this paper is that voxel selection is performed separately for each condition of interest (situation, state, etc.), and later conjunctions of the voxel masks (e.g., conjunction of situation and state voxel masks) are found during situation-state analysis. This violates the assumption of the reliability-based voxel selection method because the selected feature masks represent reliable voxels unique to their own condition (e.g., situation), which is largely non-independent from the main experimental question.

The authors justify the use of overlapping/conjunction masks by stating that they would like to observe the overlapping representation between the conditions to see how they sum up. However, this justification itself suggests that the two analyses are non-independent.

Another issue with this paper is that the same dataset is used to define reliable voxels and to analyze their responses. The reliability-based voxel selection paper explicitly describes that approaches aiming to analyze neural similarity from multi-voxel patterns should not use the same data to define reliable voxels and analyze their responses. Given this problem, the authors should consider using a different dataset to define reliable voxels and then apply the masks to a new dataset (e.g., left-out-half) to analyze the data on the main research question.

Second, it is unclear why and how the authors set up two separate GLMs. The authors take a very unconventional approach to GLM by separating conditions into two GLMs, such as one including the situation in the context of situation-action trials and another including actions in the context of situation-action trials. The authors use a boxcar regressor for the entire duration of the trials using the HRF, not FIR, and it is not clear how the situation in situation-action trials and action in situation-action trials are separated. Further clarification and justification are needed.

There are other methodological details that should be addressed. For example, were voxel timecourses z-transformed within a run? Were they corrected for temporal autocorrelations? These two are conventional approaches used when applying the reliability-based voxel selection method or conducting multivoxel pattern analyses but didn't seem to be included in the paper. Please clarify and justify whether z-scoring or correction for temporal autocorrelation was included.

The authors use Pearson correlation instead of rank correlation when comparing between neural data (e.g., summed data vs. situation). It is generally recommended to use rank correlation (e.g., Kendall's tau A) when comparing representational spaces. Is there a specific reason for choosing Pearson correlation?

Minor points:

When analyzing the summed-up representation, is it necessary for the representations to spatially overlap to be summed up? Can each representation exist in spatially separate voxels of the brain but still be summed up by another region?

typo: page 14 line 3: the result suggests that people's understanding of both the external and external influences

In the intro, the reference to Bill Nye didn't work very well for me. It may be more helpful to put this example in a more general way.

Response to reviewers. Reviewers' comments are *italicized*. Comments have been renumbered for clarity. All page numbers refer to the revised manuscript.

Reviewer #1

- 1. The RSA results nicely indicate that there is a correlation between similarity matrices (between action and states) with corresponding ratings of the likelihood of co-occurrence between these stimuli. This indicates that the degree to which the representation of action A is similar to state S depends on the likelihood that A occurs in S. Therefore, if S affords three actions (A1, A2 and A3), it is potentially correlated with all three actions (though the strength of the correlation varies between actions, obviously depending on the cooccurrence). This means that it is very likely that S is also correlated/similar to A1+A2+A3 (if B is correlated with A1, A2, A3, it is highly likely that it is correlated with their sum too). The authors have done a similar analysis (though not using RSA) and have concluded that S represents the summed actions it affords. Without doing formal model comparison, it is impossible to judge whether the representation of S is only similar to the weighted sum of the action or only one of them (e.g., the most likely one). There is a recent development in testing different models of representational geometry (see <https://arxiv.org/abs/2112.09200>). I highly recommend the authors to use this tool to test different models (i.e., the weighted sum and individual action models). The toolbox can be used to estimate the weight of different RDMs (the actions afforded by a state or situation in). The authors can test whether the weighted model is the best model and whether the weights estimated by the representational model is correlated with the reported probability of the occurrence. I believe without such a robust model comparison, the authors cannot claim that the representational geometry is “best” explained by the sum of the actions.*

We thank the reviewer for this question, as we think it raises an interesting issue that we had not fully explored in our initial submission. The critical question is whether actions are weighted when they sum to situations or mental states, or whether an alternative model best predicts the relation between them. Following from our previous work, our preregistered hypotheses assumed that the summation would follow a linear weighted average, such that the least afforded actions would take on weights close to zero, and more afforded actions would take on weights linearly proportional to their co-occurrence with the situation/state in question.

However, this is not the only weighting pattern that could make sense. For example, the brain could place a disproportional amount of weight on the most likely actions. We take the reviewer's suggestion of a “most likely” action model as a special limiting case of this hypothesis, in which all the weight is placed on the single most likely action to occur in any situation/state. Another alternative that relaxes the assumptions of our original weighting function is the possibility that some of the weights are negative. For example, under this account, the neural representation of a situation like “dinner at a restaurant” might include a high

positive weight for “eating” and a high negative weight (rather than near-zero weight) for “skiing.”

To investigate these possibilities, we conducted new representational similarity analyses. The goal of these analyses was twofold. The first, narrower goal was to test whether situation/mental state patterns were the sum of multiple actions rather than just a single action, as the reviewer suggests. The second, broader goal was to estimate the overall shape of the optimal co-occurrence weighting function. These analyses offer great new insights into the construction of social representations in the brain.

Our specific analytic approach to achieve these goals differed slightly from the model comparison approach the reviewer suggested, but still allowed us to test their specific proposal. We first binarized the co-occurrence matrix from our original analysis, such that there was a 1 in each row, corresponding to the most likely action to occur in that situation/state. We repeated this process 59 more times, except instead of the first most likely action, we created a binarized co-occurrence matrix for the second most likely action, the third, and so on. We then regressed the neural RDMs (situation-action or state-action) onto the corresponding set of 60 model RDMs in a multiple regression RSA. This allowed us to test the extent to which the most predictive action, or any of the Nth most 1st to 60th, contributes to the neural representations of situations or states. We tested the statistical significance of all 60 model RDMs across participants using the same approach we applied in the main RSAs we originally reported.

The results of this analysis (Figure S7, reproduced below) addressed the two goals described above. First, as the reviewer suggests, the most likely action was indeed predictive of both situations and states. However, this top action was *not* sufficient to account for the overall effect of co-occurrences on neural pattern similarity. For both situations and states, multiple actions other than the most likely one, were also significant predictors of the neural RDM. This further corroborates our original hypothesis that situations and mental states are sums of multiple action affordances.

Second, the results offer a precise way of describing the relation between actions and situations/states. By fitting a loess curve to the effect sizes of the 60 model RDMs, we obtained a smooth estimate of the shape of the optimal co-occurrence weighting function. The shape of this function was similar across both the situation-action and state-action cases. In both cases, it was nonlinear. Broadly consistent with the reviewer’s prediction, we observed that highly likely actions were disproportionately highly weighted relative to what a linear weighting would predict. Additionally, some of the weights were significantly negative. This suggests that situations/states are defined not only by the sum of actions they *do* afford, but also in the negative by the sum of actions they *don’t* afford. This analysis thus sheds new light onto the way the mind represents situations and states based on the actions they afford.

We believe these findings have substantially enriched the paper. It rules out the alternative model that the most likely co-occurrences (or any of the N^{th} most likely) could solely explain the original pattern summation effects. It also provides a more precise estimate of how co-occurrences are weighted when the brain sums up action representations to form situation and mental state representations. We include the details of the new analyses in the supplementary materials (p. 45-46), including the figure below (Figure S7). We also consider the implications of these results in the discussion on pages 17-18.

Figure S7. Representational similarity analyses estimate how much the 1st through 60th most likely action in each (A) mental state and (B) situation contributes to explaining pattern similarity. The results indicate that mental states and situations are nonlinear functions of the actions they afford, and are defined not only by the actions they do afford, but also in the negative by those they do not. The x-axes indicate the average co-occurrence ratings for the least likely action (left-most point) through the most likely action (right-most point). The y-axes indicate the mean neural pattern similarity between states/situations and each action rank. Red points indicate action ranks that significantly predict neural similarity ($p < .05$, controlling for multiple comparisons via maximal statistic permutation testing). Loess curves with 95% Bonferroni-corrected confidence intervals provide a smooth estimate of the optimal weighting of action patterns to reconstruct state/situation patterns.

2. *In many situations, we know someone's mental state (like anger) but we do not know what they might do when they are in that state. What does the framework suggested by this study predict for these situations?*

Participant ratings do indeed indicate that some mental states make more precise predictions about actions, whereas others make more diffuse predictions. In real life, people generally have access to more sources of information than just a mental state upon which to base an action prediction. By combining these multiple sources of information, they are likely to be able to disambiguate otherwise imprecise mappings between states and actions. For example, knowing whether someone is around other people or not would help bound the set of actions they are likely to take when angry. We discuss this issue further on page 16.

3. *The authors report that “A set of 570 voxels were shared between the situation and action maps, 668 voxels were shared between the mental state and action maps, and 1545 voxels were shared between the situation and mental state maps. The existence of this spatial overlap provides an initial indication that neural representations of situations, states, and actions may be relate”. I could not find any information about the spatial map of these voxels. Can the authors report such a map? It can be useful for understanding the computational role of some of the key brain regions in social cognition. For example, Roumazeilles et al. Science Advance, 2021 shows that an area in mid-STS (an area most similar to human TPJ) responds to social prediction error. Did the authors find any voxels in this area of the human brain?*

The situation-action overlap map was shown in Figure 5 of the original submission, but we now include all three maps together in an additional supplemental Figure S5. As this new figure shows, the STS is indeed heavily implicated in the overlap between situations, states, and action representation. This includes a pSTS/TPJ region, a (human) mid-STS region, and an anterior temporal lobe region. We report where we see neural overlap across representations on page 10.

4. *Minor: It would be very useful if the authors relocate Figure 5 to the result section where they introduce their methods*

We agree and have relocated this figure. The figures have been renumbered in the revised manuscript as a result.

Reviewer #2

5. *The authors are performing voxel feature selection based on the reliability-based feature selection method. The critical advantage of the reliability-based voxel selection method (Tarhan & Konkle) is that experimenters can select reliable voxels independent of main condition effects and data. For instance, when selecting voxels for object size analysis (comparing big vs. small objects), all object trials (e.g., 100 objects) should be considered, regardless of object size. Voxels with high reliability (within > across activity for each trial type) can then*

be selected. This allows the selection of voxels independent of the question of interest, which is about the size of objects. The problem with the voxel feature selection approach used in this paper is that voxel selection is performed separately for each condition of interest (situation, state, etc.), and later conjunctions of the voxel masks (e.g., conjunction of situation and state voxel masks) are found during situation-state analysis. This violates the assumption of the reliability-based voxel selection method because the selected feature masks represent reliable voxels unique to their own condition (e.g., situation), which is largely non-independent from the main experimental question. The authors justify the use of overlapping/conjunction masks by stating that they would like to observe the overlapping representation between the conditions to see how they sum up. However, this justification itself suggests that the two analyses are non-independent.

The approach outlined by Tarhan and Konkle (2020) rests on an assumption: that the stimuli/conditions of the experiment belong to the same psychological domain. This seems like a reasonable assumption in the case they consider: all their stimuli are objects. Although two objects might differ on one dimension, such as real-world size, they might be similar on another, such as animacy. For example, a mouse and an elephant differ greatly in size, but both are animate. This makes it reasonable to conduct reliability-based feature selection across all of them.

However, not all possible stimuli/experimental conditions belong to the same psychological domain, nor do all parts of the brain process all classes of stimuli. In the present case, we examined three quite different types of stimuli: situations, mental states, and actions. As Tarhan and Konkle themselves show, the brain regions that reliability-based feature selection implicates in action perception are quite different from those typically implicated in representing mental states. For example, regions like the temporoparietal junction and medial prefrontal cortex – long implicated in theory of mind – are conspicuously absent from the regions implicated in action perception by their data. Thus, it seemed to us quite unsafe to assume that the three types of stimuli we investigate exist within the same representational space or are processed by the same set of brain regions.

Rather than making this unwarranted assumption, we adapted our use of reliability-based feature selection to test this assumption. By performing feature selection separately on each of these domains, we empirically tested whether they were represented by any shared neural substrates. Testing this assumption first reflects an even more conservative approach, as it opens us up to the possibility that there is no neural overlap, and thus, no possibility to test if actions sum to either states or representations. This conservative test would not have been possible had we performed reliability-based feature selection across all three stimuli types at once. If we had done that, we would have been guaranteed by the nature of the algorithm to select some non-zero set of voxels. That non-zero set of selected voxels would reflect regions that reliably represent situations *or* mental

states *or* actions. There would be no guarantee that any of the voxels selected actually represents all three (or even two) of these types of stimuli. The presence of reliable representations of at least two classes of stimuli in the same set of voxels is a necessary prerequisite for representations of one class to sum up to representations in another class. Performing reliability-based feature selection across all stimuli would have prevented us from verifying this key assumption underlying our subsequent pattern analyses. We now explain this logic more explicitly on pages 32-33.

The reviewer is correct that our main hypotheses are not independent of the feature-selection, but the dependence between them is logical, not statistical. That is, the existence of spatial overlap between situation, state, and action-representing patterns is a necessary condition for it to make sense to add up patterns of one type to reconstruct patterns of another. This form of dependence is not problematic. However, the way that the term non-independence is typically used in the fMRI literature refers to statistical non-independence, in the sense of circularity. For example, if a univariate contrast is used to define an ROI, and then the same contrast is tested with the same data in the same ROI, the effect will be inappropriately inflated. The present study does not suffer from this type of statistical non-independence between the feature selection and the hypothesis tests. The results of the feature-selection (i.e., the existence of overlapping voxels) reveal that the main hypotheses are possible, but do not guarantee that they are true, nor does it bias us in the direction of corroborating them. We explain why in the next response below.

6. *Another issue with this paper is that the same dataset is used to define reliable voxels and to analyze their responses. The reliability-based voxel selection paper explicitly describes that approaches aiming to analyze neural similarity from multi-voxel patterns should not use the same data to define reliable voxels and analyze their responses. Given this problem, the authors should consider using a different dataset to define reliable voxels and then apply the masks to a new dataset (e.g., left-out-half) to analyze the data on the main research question.*

The reviewer is correct that Tarhan and Konkle (2020) recommend using different data to perform feature selection and analyze the data. However, this recommendation applies specifically to using reliability-based feature selection for decoding models or representational similarity analyses in which the *conditions themselves* are the variable of interest. For example, the decoding model presented by Tarhan and Konkle reflects the pairwise accuracy of classifying whether a given pattern of brain activity corresponds to one action vs. another. Since these actions are both the target of the decoding, and the conditions over which reliability was calculated during the feature selection procedure, this decoding would indeed yield biased (i.e., circular) results if it was performed on the same data as the feature selection.

In their prior empirical paper on this same dataset, Tarhan and Konkle (2019) also fit a voxelwise encoding model. This model predicts the brain activity associated with the actions using a separate set of independent variables, such as how social the action was, or what part of the body was involved. If these independent variables had been used for feature selection (e.g., choosing voxels were activity correlated with sociality), this would have been a classic case of double-dipping. However, since these independent variables were not involved in the reliability-based feature selection, the authors could – and did – perform both the feature selection and the encoding modeling using the same data.

The analyses in the present paper are conceptually similar to this latter encoding model case. We are not trying to decode the situations, states, or actions from patterns of brain activity. Rather, we are testing whether the co-occurrence ratings provide appropriate weights to sum up patterns of one class to reconstruct patterns of another class, or in the RSA, whether these ratings are correlated with neural pattern similarity. Thus, whether our hypotheses are validated hinges on the co-occurrence ratings. Since these ratings were not involved in the feature selection process – and importantly, were not used to select voxels that favored our hypotheses – there is no circularity bias in our results, nor a need to use separate data for feature selection and hypothesis testing.

This is not the first time that concerns around circularity have arisen with respect to reliability-based feature selection. It is a rather nuanced issue, easily susceptible to misunderstanding, which is why of us wrote about this exact issue at length in 2015: <http://markallenthornton.com/blog/fmri-reliability/> The linked post contains a detailed description of this issue, as well as code for simulations that back up its assertions. Reliability-based feature selection on the same data actually makes estimates of effect sizes *more* accurate, not less. For this reason, we have retained the original approach in the paper. However, we have now clarified the validity of this approach on page 34.

7. *It is unclear why and how the authors set up two separate GLMs. The authors take a very unconventional approach to GLM by separating conditions into two GLMs, such as one including the situation in the context of situation-action trials and another including actions in the context of situation-action trials. The authors use a boxcar regressor for the entire duration of the trials using the HRF, not FIR, and it is not clear how the situation in situation-action trials and action in situation-action trials are separated. Further clarification and justification are needed.*

Using two separate GLMs was necessary for model specification purposes. As the reviewer notes, due to the nature of the task, each trial falls into two different conditions. For example, a situation-action trial would feature both 1 of the 60 situations, and 1 of the 60 actions – with participants rating how likely that action was to occur in that situation. However, if two regressors modeled each trial, it would be impossible to estimate the model parameters because each regressor

would be a linear combination of another set of regressors. There are different approaches one might apply to address this issue, but using two GLMs was the most straightforward and consistent with our preregistered plans. We have clarified the reasons behind this analytic decision on page 30.

We indeed modeled the data using boxcar regressors convolved with a canonical hemodynamic response function instead of a finite impulse response model. The FIR approach certainly has some advantages – particularly when one is interested in estimating the temporal profile of responses – but these benefits are not relevant to the goals of the current project. Since the boxcar + HRF approach is quite standard in the literature, and more common than FIR, and it was our preregistered approach, we have continued using it in the revision.

8. *There are other methodological details that should be addressed. For example, were voxel timecourses z-transformed within a run? Were they corrected for temporal autocorrelations? These two are conventional approaches used when applying the reliability-based voxel selection method or conducting multivoxel pattern analyses but didn't seem to be included in the paper. Please clarify and justify whether z-scoring or correction for temporal autocorrelation was included.*

We are happy to provide more details on our methodological approach. First, we did not z-transform voxel time courses within a run. However, we did model all runs in the same GLM, which achieves the same effect, as we included run means as confound regressors. Second, we did not correct for temporal autocorrelations. This step is not necessary unless one wishes to compute the statistical significance of GLM parameters at an individual subject level. That is because the dependence induced by temporal autocorrelation biases the standard errors of the GLM parameters (i.e., by making the degrees of freedom inappropriately large) but does not bias the parameters (betas) themselves. Given that we do not compute statistical significance on these betas, or indeed at the voxelwise level at all, this step is not necessary for our approach.

Since these steps are not necessarily standard preprocessing for fMRI data, we think it might be more confusing than clarifying to mention in the paper that we did not apply them. That said, we would be happy to discuss their absence if requested.

We adapted our modeling approach from prior published work, specifically, Tarhan and Konkle. That prior paper did not justify these preprocessing steps in their methods paper, nor in the earlier empirical paper upon which it is based. We speculate that the z-scoring of each run was performed to adjust for differences in mean across runs (since not all of their runs were in the same GLM). Thus, we included an appropriate approximation of this step. The method of correction for temporal autocorrelations is not specified in the paper, making it difficult for us to emulate.

9. *The authors use Pearson correlation instead of rank correlation when comparing between neural data (e.g., summed data vs. situation). It is generally recommended to use rank correlation (e.g., Kendall's tau A) when comparing representational spaces. Is there a specific reason for choosing Pearson correlation?*

We thank the reviewer for this suggestion. The revised manuscript includes a supplementary analysis showing that our results replicate when using Kendall's tau (p. 44-45). For this purpose, we used tau-B rather than tau-A. Kendall's tau A is recommended when comparing categorical and continuous models, because tau B can overestimate the performance of categorical models. However, such a comparison is not a part of the RSAs we reported previously – each neural RDM is correlated with just one neural RDM in the main analyses. As a result, tau A and tau B would both be suitable choices here. However, tau A is computationally inefficient on RDMs of the size and number we use here, leading us to use the faster tau B instead.

10. *When analyzing the summed-up representation, is it necessary for the representations to spatially overlap to be summed up? Can each representation exist in spatially separate voxels of the brain but still be summed up by another region?*

Yes, it is necessary for the representations to spatially overlap to test the weighted sum model. It is unclear how we could test if the sum of action representations from one region matched state or situation representations from a different region. Comparing patterns requires correlating their values across the same set of voxels. The correlation between unmatched voxels would be meaningless. It is possible that a pattern in one region represents some transformation of a pattern in another region, but examining such relationships is beyond the scope of the present investigation.

11. *typo: page 14 line 3: the result suggests that people's understanding of both the external and external influences*

Thank you, we have fixed this typo.

12. *In the intro, the reference to Bill Nye didn't work very well for me. It may be more helpful to put this example in a more general way.*

Thank you for pointing out this problematic example. We have edited the example to improve its clarity.

Reviewer #1 (Remarks to the Author):

The authors tried to address my concerns. Although, most of their replies are convincing, their reply to my main concern about model comparison is not sufficient. In the following I will explain why:

The authors suggest a very specific model: that is "The sum". How do we know this is the best model? Even if all actions are significant, why is the "SUM" the best model? The authors showed that including all actions in their model explains extra variance. I did not seem to be able to convince the authors that model comparison is absolutely necessary to make this claim. So, I am going to use linear regression as an example.

I hope the authors agree that, in linear regression, adding an extra regressor with a significant beta value does not justify its inclusion. There are tools for model comparison which is the only way to justify inclusion of a regressor. How is this case different from linear regression? Inclusion of all actions to explain neural representational geometry can only be justified by using model comparison. I think to make this claim, the authors must show that the model they propose is the best model, by using reliable model comparison tools.

In addition, they should also do the reverse of what they did. They should compute the neural weight between each action and each state and show that these weights are correlated with the probability of their cooccurrence.

Reviewer #2 (Remarks to the Author):

In my previous review, I had significant concerns around the issue of non-independence in voxel selection and analysis. In their response, the authors have undertaken a comprehensive explanation addressing how the three types of stimuli used in their experiment logically overlap but with no statistical non-independence. Specifically, their method is not based on a decoding approach, instead adopting a voxel-based encoding method to estimate the weight of an independent third variable, the co-occurrence ratings.

I find myself convinced by these arguments and appreciate the detailed explanation and added description on page 34 of their manuscript. It is helpful for readers that the authors specifically describe how the current paper is exempt from the circularity in the reliability-based voxel selection method. The other clarifications for my other methodology comments are clear and I have no further concerns.

Response to reviewers. Reviewers' comments are *italicized*. Comments have been renumbered for clarity. All page numbers refer to the revised manuscript.

Reviewer #1

- 1. The authors tried to address my concerns. Although, most of their replies are convincing, their reply to my main concern about model comparison is not sufficient. In the following I will explain why: The authors suggest a very specific model: that is "The sum". How do we know this is the best model? Even if all actions are significant, why is the "SUM" the best model? The authors showed that including all actions in their model explains extra variance. I did not seem to be able to convince the authors that model comparison is absolutely necessary to make this claim. So, I am going to use linear regression as an example.*

I hope the authors agree that, in linear regression, adding an extra regressor with a significant beta value does not justify its inclusion. There are tools for model comparison which is the only way to justify inclusion of a regressor. How is this case different from linear regression? Inclusion of all actions to explain neural representational geometry can only be justified by using model comparison. I think to make this claim, the authors must show that the model they propose is the best model, by using reliable model comparison tools.

In addition, they should also do the reverse of what they did. They should compute the neural weight between each action and each state and show that these weights are correlated with the probability of their cooccurrence.

We are glad that we have addressed most of the points that the reviewer raised in their previous comments, and we appreciate their patience as we work to resolve this final concern.

We completely agree with the reviewer's high-level comments – both now and in their previous review – regarding the importance of appropriate model comparison. Indeed, in some of our prior work (Thornton & Tamir, 2020, *Cortex*), we took an approach that was generally similar to the reviewer's original suggestions.

Unfortunately, the specific approach the reviewer suggested previously (Schütt et al., 2021) cannot be applied to the current data. Specifically, we perceive there to be two challenges to applying that method. First, that method was not designed to accommodate representational dissimilarity matrices (RDMs) that feature different conditions on the rows and columns, as do the matrices we analyze here. Second, it is not possible to generate a full model RDM for each action, because each action corresponds to only a single row or column of one of the neural RDMs. This makes it impossible to compare the individual actions to each other, or their sum, because each action makes predictions about a different part of the neural RDM.

Fortunately, there are several other methods at our disposal to address the substantive issues the reviewer raised via model comparison. To this end, we added three new model comparison analyses based on our pattern summation analyses, rather than representation similarity analysis:

- i) Our first analysis in this vein directly compares situation/state patterns to two different models thereof. The first model consists of the affordance-weighted sums of action patterns which we examined before. The second model consists of the patterns corresponding to whatever action is most likely in each situation/state, as the reviewer suggested. We compute the RMSE between each situation/state pattern, and the two respective reconstructions. We then test which of the two reconstructions is more accurate (i.e., lower RMSE). We find that the summed actions significantly outperform the single most likely action in this respect. This provides direct evidence in favor of the summed action model over the alternative suggested by the reviewer.
- ii) The second analysis we performed compares the two models to each other, and also examines whether they explain unique information. To this end, we fit three regressions: we regress each situation/state pattern onto either the corresponding summed action model, single most likely action model, or both of these models simultaneously. We repeat this process for every situation/state, within each participant. From each regression, we extract AIC as a measure of model performance. We average AICs across situations/states, and then compare the mean AICs of the three regressions across participants via pairwise paired t-tests. The results of this analysis show that the summed action model either significantly outperforms, or is not significantly different from, the single action model. The regression containing both the summed and single action models achieves significantly lower AICs (i.e., better performance) than the other two. This set of results further corroborates the hypothesis that situation/state representations are the sums of multiple action representations. However, it also suggests that highly likely actions are *disproportionately* weighted in these sums. This latter result is consistent with the RSA we introduced in the previous round of review, where we showed the optimal weighting function was nonlinear, featuring disproportionately high weights for the most probable actions, and negative weights for unlikely actions.
- iii) The final analysis we conducted to address this point consisted of cross-validated model selection. This analysis compared all possible linear combinations of actions to find the optimal combination for reconstructing each situation/state. Specifically, we use LASSO regression to decompose each situation/state pattern into a combination of action patterns. Using cross-validation, we were able to select the optimal LASSO penalty value for each fold of this regression. We then examined the version of the LASSO with this lambda to determine how many non-zero coefficients (besides the intercept) it had. This provides us with an estimate of how

many actions optimally contribute to each situation or mental state. Since each situation/state is analyzed separately in this procedure, they can be regarded as statistically independent. This allows us to perform statistical inference over the mean number of actions that contributes to each situation/state. We find that 44% of actions contribute to each situation and 40% of actions contribute to each mental state. This outcome clearly shows that the optimal reconstruction of situation and mental state representations comprises many actions, not just one (e.g., the most likely). However, in itself, this analysis does not show that the contributions of these actions are in proportion to the co-occurrences between the situations/states and the actions. To test this, we followed the reviewer's suggestion by performing a variation on our original RSAs in which the neural RDMs were composed of optimized LASSO regression weights instead of zero-order correlations between situation/state patterns and action patterns. We found statistically significant correlations between the situation-action neural weight matrix and affordance matrix, and between the mental state-action neural weight matrix and affordance matrix.

The methods and results for these analyses are reported on pages 13-15 and 39-41 of the revised manuscript, and pages 1-4 of the Supplementary Materials. Together, the results of these model comparisons show that situation and mental state representations are indeed comprised of affordance weighted sums of multiple action representations. We appreciate the reviewer's encouragement to develop these new analyses, as we believe they add substantially to the conclusions that can be drawn from this work.

Reviewer #1 (Remarks to the Author):

The authors have addressed my remaining concern. I would like to congratulate the authors for their nice study. Looking forward to seeing it out.